# A Scoping Literature Review of the Relation between Nutrition and ASD Symptoms in Children

**DOI:** 10.3390/nu14071389

**Published:** 2022-03-26

**Authors:** Inge van der Wurff, Anke Oenema, Dennis de Ruijter, Claudia Vingerhoets, Thérèse van Amelsvoort, Bart Rutten, Sandra Mulkens, Sebastian Köhler, Annemie Schols, Renate de Groot

**Affiliations:** 1Health Psychology, Faculty of Psychology, Open University of the Netherlands, 6419 AT Heerlen, The Netherlands; 2Conditions for Lifelong Learning, Faculty of Educational Sciences, Open University of the Netherlands, 6419 AT Heerlen, The Netherlands; renate.degroot@ou.nl; 3Department of Health Promotion, Maastricht University, 6200 MD Maastricht, The Netherlands; a.oenema@maastrichtuniversity.nl (A.O.); d.deruijter@maastrichtuniversity.nl (D.d.R.); 4School of Nutrition and Translational Research in Metabolism (NUTRIM), Maastricht University, 6200 MD Maastricht, The Netherlands; a.schols@maastrichtuniversity.nl; 5Department of Psychiatry and Neuropsychology, Maastricht University, 6200 MD Maastricht, The Netherlands; claudia.vingerhoets@maastrichtuniversity.nl (C.V.); t.vanamelsvoort@maastrichtuniversity.nl (T.v.A.); b.rutten@maastrichtuniversity.nl (B.R.); sandra.mulkens@mumc.nl (S.M.); s.koehler@maastrichtuniversity.nl (S.K.); 6School for Mental Health and Neuroscience (MHeNs), Maastricht University, 6200 MD Maastricht, The Netherlands; 7Department of Clinical Psychological Science, Maastricht University, 6200 MD Maastricht, The Netherlands; 8SeysCentra, 6581 TE Malden, The Netherlands; 9Department of Respiratory Medicine, Maastricht University, 6202 AZ Maastricht, The Netherlands

**Keywords:** autism spectrum disorder, nutrients, diet, dietary pattern, children

## Abstract

Autism spectrum disorder (ASD) is characterized by impairments in social interaction, communication skills, and repetitive and restrictive behaviors and interests. Even though there is a biological basis for an effect of specific nutrition factors on ASD symptoms and there is scientific literature available on this relationship, whether nutrition factors could play a role in ASD treatment is unclear. The goal of the current literature review was to summarize the available scientific literature on the relation between nutrition and autism spectrum disorder (ASD) symptoms in childhood, and to formulate practical dietary guidelines. A comprehensive search strategy including terms for ASD, nutrition factors (therapeutic diets, dietary patterns, specific food products, fatty acids and micronutrients) and childhood was developed and executed in six literature databases (Cinahl, Cochrane, Ovid Embase, PsycInfo, PubMed and Web of Science). Data from meta-analyses, systematic reviews and original studies were qualitatively summarized. A total of 5 meta-analyses, 29 systematic reviews and 27 original studies were retrieved that focused on therapeutic diets, specific food products, fatty acids and micronutrients and ASD symptoms during childhood. Results of the available studies were sparse and inconclusive, and hence, no firm conclusions could be drawn. There is currently insufficient evidence for a relation between nutrition and ASD symptoms in childhood, making it impossible to provide practical nutrition guidelines; more methodological sound research is needed.

## 1. Introduction

Autism spectrum disorder (ASD) is a neurodevelopmental disorder characterized by three core behavioral symptoms: impairment in social interaction skills, impairment in communication, and repetitive and restrictive behaviors and interests [1]. Estimations of the prevalence of ASD vary and is nowadays estimated to be globally around 0.06–1.4% [2]. The impact of ASD on patients’ quality of life is high. ASD is for example related to poor educational and employment outcomes, anxiety, depressive symptoms, sleep problems, loneliness, isolation and limited societal participation [3,4]. Additionally, there are large societal and financial consequences related to ASD, with average costs related to support an individual with ASD estimated to be up to 3.2 million dollars across the lifetime of a person with ASD in the United States and United Kingdom [5,6].

The exact etiology of ASD is unknown, but it has been suggested to be multifactorial and possibly heterogeneous (i.e., symptoms could be explained by multiple etiologies with overlapping symptoms) [7,8,9]. Hence, genetic, neurologic, metabolic and immunologic factors have been implicated to be involved in the etiology [10]. Additionally, it has been hypothesized that the interaction between genes and environmental factors may also play a role [11].

Unfortunately, the current pharmacological treatment options for individuals with ASD are limited, there are psychological behavioral treatments, however these might not be available, feasible or effective for all persons with ASD [12,13]. For example, early intensive behavioral interventions show vast inter-individual differences in the effectiveness, need to start early, ideally before age four, require a vast time investment of both the child with ASD and the parent (i.e., 25 h per week or more) and the expense of such treatments might not make them available for all those with ASD. In this context, nutrition has been increasingly receiving attention as potential treatment option. ASD is known to be associated with biological processes on which nutrition can have an effect such as immune dysregulation, inflammation, impaired mitochondrial function, redox regulation problems and oxidative stress [14,15]. Moreover, children with ASD often have restrictive dietary patterns [16] and nutritional deficiencies are common in those with ASD [17,18], making balanced nutrition even more important for children with ASD.

Furthermore, adequate nutrition is important for brain development, functioning, and maintaining health in general [19]. Specifically, in periods of profound brain development such as the prenatal period, in childhood and in adolescence, the brain is vulnerable to the effects of nutrient deficiency, but might also be susceptible to positive effects of nutrients [19]. Many nutrients are important for both the structural integrity of the brain and functional processes such as synaptogenesis, neurotransmitter synthesis, DNA methylation and membrane functioning [19].

Considering the importance of nutrition for many bodily processes, brain development, and functioning, the rather limited effective treatment options for those with ASD, the putative benefit of nutrition on ASD, and the observed food selectivity and nutritional deficiencies in those with ASD, it might not come as a surprise that the scientific interest concerning the potential effect of nutrition on ASD has risen. Past research has investigated the effects of numerous diets and nutrients on ASD symptoms; for example the gluten and/or casein free diet, omega-3 fatty acids [20], and vitamin D [21].

However, to the best of our knowledge, no scoping review summarizing studies investigating the relation between a wide range of nutrition factors and ASD symptoms in children is available, which limits the translation of this knowledge to application in practical care. Therefore, our goal is to summarize the scientific literature, regarding the relation between nutrition and ASD symptoms in childhood by means of a comprehensive scoping literature review; in doing so we want to provide an extensive overview about what nutrition factors have been studied in relation to ASD symptoms in childhood, what conclusions can be drawn and what gaps in the knowledge base are. Another aim is to to develop practical nutrition guidelines.

## 2. Materials and Methods

This review was part of a larger knowledge synthesis in which the relationships between nutrition and mental health (problems) were investigated in a lifetime perspective [22]. In the current manuscript the focus is on the relation between dietary patterns, therapeutic diets, specific food products, and nutrients (i.e., micro and macro nutrients) during childhood and ASD symptoms during childhood (here defined as children aged 0 to 12 years). Dietary pattern was defined as the habitual total food intake, while a therapeutic diet was defined as a specific diet prescribed for the treatment of ASD, e.g., the gluten and casein free diet.

To execute the knowledge synthesis, a comprehensive search strategy was developed which included search terms related to (1) nutritional factors (e.g., dietary patterns, therapeutic diet, and nutrients), (2) mental health problems specific for the life phase (i.e., for the current review: ASD), (3) the specific age group (i.e., children). The overall childhood search strategy can be found in the Appendix A. The search strategy was executed in the following databases: Cinahl, Cochrane, Ovid Embase, PsycInfo, PubMed, and Web of Science. No restrictions on publication year were entered in any of the databases. The overall search strategy was adapted for each of the databases searched.

The search strategy was executed in July 2018 and updated in April 2020. All found articles were first title/abstract screened, and then full text screened by the first author. Title/abstract screening was checked by a second screener and discrepancies were solved by discussion.

If multiple meta-analyses or systematic reviews were available, a review of reviews was executed. This review of reviews was supplemented with original articles not included in any of the meta-analyses or systematic reviews. If no meta-analyses or systematic reviews were available, the results of the original studies were summarized.

### 2.1. Inclusion and Exclusion Criteria

Studies (i.e., studies in meta-analyses, studies in systematic reviews and original studies) were included in this review if (1) they reported on ASD or ASD symptoms or characteristics; (2) the ASD outcome measure used was either an ASD specific clinical-diagnostic instrument or a parent or teacher-rated instrument related to ASD; (3) dietary patterns, therapeutic diet, specific food product, or nutrient intake were measured with a quantitative instrument OR the dietary pattern, diet, or nutrient intake was manipulated by diet intervention, dietary advise or supplementation; (4) the study included children aged 0–12 years, or the average age of the included participants was 12 years or younger; (5) the study was reported in English, Dutch or German; and (6) the manuscript was a published meta-analysis, systematic review, or an original study not included in the meta-analyses or systematic reviews included in the current review.

Studies (i.e., studies in meta-analyses, studies in systematic reviews and original studies) were excluded if the study (1) was a single-case study; (2) a cross-sectional study reporting on nutrition intake (e.g., 24 h diet recall, food frequency questionnaires or diet diary) or nutritional status (e.g., blood values) only of children with ASD (3) a case-control study in which the nutrition intake (e.g., 24 h diet recall, food frequency questionnaires or diet diary) or nutritional status (e.g., blood values) only of children with ASD was compared to children without ASD; (4) reported on pre- or probiotics, food coloring, preservatives, alcohol consumption, caffeine consumption, or herbal supplements, as these are not considered as normal elements of a regular dietary pattern for children.

Note that in some meta-analyses and systematic reviews article were included that did not meet in- and exclusion criteria of this review, these were not mentioned in the study design(s) column of the meta-analyses/systematic review result tables nor counted in the total of included studies in the text. However, they might be present in meta-analyses results, if this is the case this is noted via a superscript next to the study design cell in the meta-analyses/systematic review result tables.

### 2.2. Data Extraction

For the review of reviews, the following data were extracted from each included meta-analysis/systematic review: type of study (meta-analysis, systematic review, combination), design of studies included, number of studies included relevant for our review, characteristics of included participants (i.e., age range and specifics regarding ASD diagnosis), nutrition component or described intervention, name of the first author, publication year and results. Regarding the results for meta-analyses, standardized effect sizes (i.e., odds ratios, standardized mean difference or Hedges G) and confidence intervals were extracted. For systematic reviews the number of studies with a significant effect or association, relative risk, odds ratio or effect size were abstracted. When a systematic review described multiple exposures (for example micronutrients and pollutants) only those results relevant for the current review were extracted. Additionally, some meta-analyses and systematic reviews reported on multiple dietary patterns, therapeutic diet or nutrient. In the current review, the results are discussed per dietary pattern, therapeutic diet or nutrient. Therefore, it can be the case that a meta-analysis or systematic review is discussed multiple times. It is also important to note that some original studies were included in a multitude of meta-analyses and/or systematic reviews and that the results of this original study are thus ‘counted’ more than once in the weight of evidence (also see analyses). Characteristics of the meta-analysis and systematic reviews are presented in the tables and summarized in the text.

For the update of review of reviews and the review of original studies the following data were extracted from each included original study: design, number and characteristics of participants, the ASD measuring instrument, the nutritional component or intervention, the treatment in the control group, name of the first author, year of publication and results. Characteristics of the original studies and their results were presented in tables and summarized in the text. Note that an original study is defined as a study which was not included in any of the meta-analyses or systematic reviews.

### 2.3. Analyses

For the review of reviews, the results of the meta-analyses and systematic reviews were integrated in a self-developed qualitative way. If there were multiple meta-analyses on the same topic, we described how many of those meta-analyses found an effect in the same direction and what the size of the effect or the association was. For systematic reviews we examined the direction and consistency of the findings of the systematic reviews, and based upon this, we made a final assessment. An effect or association was considered significant if 2/3 or more of the included studies found an effect or association in the same direction (i.e., beneficial (+)/no effect (0)/unfavorable (−)) and if there were at least five studies in which the effect/association was studied. If the results in a systematic review or meta-analyses were mixed (i.e., <2/3 studies in the same direction), and there were at least five studies in which the effect/association was studied, this was indicated with +/−. If the effect/association was studied < 5 studies no conclusion was drawn.

For the update of review of reviews with original studies, the results of original studies were summarized qualitatively. Original studies in which the same nutritional factor (i.e., dietary pattern, specific food product, therapeutic diet or nutrient) was investigated were taken together and it was determined how many of these studies showed an effect or association in the same direction in the same manner as the review of review method.

If no meta-analyses or systematic reviews were available on a specific nutritional factor, but there were original studies available the results of the original studies were summarized in a review of original studies. The approach was equal to that of the review of reviews, an effect or association was considered significant if 2/3 or more of the included studies found an effect or association in the same direction (i.e., beneficial/neutral/unfavorable) and if there were at least five studies in which the effect/association was studied.

The quality of the original studies describing an RCT or cohort study was assessed using a quality assessment tool for quantitative studies [23]. The quality of cross-sectional studies was not assessed because this type of studies scores low on level of evidence (see Table 1). The components that were tested are: selection bias, research design, correction for confounders, blinding, validity and reliability of measuring instruments, and participant drop-out. Each component was assessed based on two questions that could be scored as “yes”, “no”, or “not clear”. The scores on the two questions resulted in a judgment per component as strong, moderate, or weak. The judgments of all components together gave a final score of the quality of the study. A study was qualified as strong when no component was assessed as weak, as moderate when one component was assessed as weak and as weak when two or more components were assessed as weak.

### 2.4. Overall Strength of Evidence

In this review, we used the following qualifications for the overall strength of evidence: very strong, strong, moderately strong, weak, very weak, insufficient evidence and no evidence. In Table 1, the different qualifications and the way in which the strength of the evidence is indicated in the text are reported. Since no criteria for strength of evidence for reviews-of-reviews and meta-analyses exist, this qualification has been developed for the knowledge synthesis that formed the basis for this review, drawing on already existing guidelines and criteria [24,25,26,27]. Note that this strength of evidence qualifications takes together the results of meta-analyses, systematic reviews and original studies. Additionally, conclusions were only drawn and the associated strength of evidence was only given when 2/3 or more of the included studies found an effect or association in the same direction and if there were at least five studies in which the effect/association was studied. If this is not the case no conclusion and no strength of evidence is given.

Based on the results of the meta-analyses, systematic reviews and original studies found, conclusions were drawn, and recommendations were made for research and clinical practice. Conclusions were only drawn if there were one or more meta-analyses/systematic reviews with a minimum of five original studies included or at least five original studies that studied a particular association or effect. Recommendations for practice were only given when there was moderately strong to very strong evidence for the effect of a nutrition factor on ASD symptoms.

## 3. Results

In total, 5 meta-analyses, 29 systematic reviews and 27 original studies were found that focused on nutrition during childhood and childhood ASD symptoms. We located studies with a focus on therapeutic diets and intake of specific food products, fatty acids and micronutrients. No meta-analyses, systematic reviews or original studies with a focus on dietary patterns for ASD symptoms were found. All meta-analyses, systematic reviews and original articles were published in English. In total, 71 unique studies were included in the meta-analyses and systematic reviews, amounting to 98 unique studies included in the current review.

Below, we first report studies on therapeutic diets and intake of specific food products, followed by studies on fatty acids and finishing with studies on micronutrients. For each nutrition factor (i.e., therapeutic diets and intake of specific food products, fatty acids and micronutrient), we first describe the number of meta-analyses, systematic reviews and original studies which were located focusing on a specific nutrition factor. Subsequently, we report the results of the meta-analyses and systematic reviews, then we report the results of the original studies and we finish with the conclusion on the effect of the specific nutrition factor on childhood ASD symptoms.

In Appendix A the original studies included per meta-analysis and systematic review are reported. Additionally, it shows the meta-analyses and/or systematic reviews in which each original study was included.

### 3.1. Therapeutic Diets

In total 18 systematic reviews focusing on therapeutic diets for children with ASD were found. In nine of these systematic reviews the focus was solely on the gluten and/or casein free diet (GFCF) [28,29,30,31,32,33,34,35,36], in three systematic reviews the focus was on the ketogenic diet [37,38,39], and in six the focus was on multiple different diets and specific food products (GFCF, ketogenic or Chanyi diet, and camel milk) [13,17,40,41,42,43] (see Table 2).

#### 3.1.1. Gluten and/or Casein Free Diet

Fifteen systematic reviews were found that included a total of 20 unique studies, in which the effect of a gluten and/or casein free diet on ASD was investigated [13,17,28,29,30,31,32,33,34,35,36,40,41,42,43] (see Table 2 and Appendix A). Approximately half of the studies included in the systematic reviews showed a beneficial effect of the GFCF diet on one or more of the ASD outcome measures which were included in the study. However, as all studies utilized different outcome measures, no specific conclusion regarding specific outcome variables can be described. Moreover, many of these studies had a weak research design (i.e., open label or single blind) and mostly included very few participants (i.e., varying between 10 and 150, with the majority of studies including fewer than 20 participants).

In addition, eight original studies were found [44,45,46,47,48,49,50,51] (see Table 3). Six of these eight original studies were intervention studies of moderate or weak quality [44,45,46,47,50,51] and two were observational studies following children who were already on a GFCF diet [48,49]. In three of the six intervention studies beneficial effects on the Childhood Autism Rating Scale (CARS) score and behavior as noted by parents were reported [47,50,51]. It is, however, important to note that these three studies did not include a control group. In the two observational studies parents reported an improvement in the behavior of their children after the start of a GFCF diet on respectively the CARS [49] and characteristic ASD behaviors and social behaviors [48]. This improvement was significantly better than those that did not eliminate gluten and casein completely [48], but not significantly better than those that were not on a GFCF diet. The chance of information bias seems high in these studies.

Overall, fewer than 2/3 of studies showed an effect in the same direction, and hence no conclusion regarding the effect of the GFCF diet for the treatment of childhood ASD symptoms can be drawn.

#### 3.1.2. Ketogenic Diet

Five systematic reviews were found in which the ketogenic diet in children with ASD was discussed [37,38,39,40,41] (see Table 2). These five systematic reviews only included a total of four unique studies. All four studies utilized the CARS (among others) as outcome measure and in all studies some, but not all participants on the ketogenic diet showed improvements on the CARS. However, a large proportion of the children either did not want to participate or dropped out during the study, making sampling and attrition bias very likely. In addition, the studies were very small (all ≤ 45 participants) and three studies did not include any control, and one study compared the ketogenic diet to the GFCF diet.

There were no original studies looking at the effect of the ketogenic diet on ASD symptoms in children.

As there were fewer than five studies available, no conclusion can be drawn regarding the effect of the ketogenic diet on ASD symptoms in children.

#### 3.1.3. Other Diets and Specific Food Products

One study that looked at the Chanyi diet was included in two systematic reviews [41,43]. This is a traditional Chinese diet that avoids products that produce “internal heat”, such as meat, fish and ginger (see Table 2). In this study significant fewer social communication problems and a significant positive change in executive control of repetitive, inflexible and hyperactive behavior was shown for the Chanyi diet group, but not in the control group in which children consumed their usual diet.

In five systematic reviews a total of two different studies looking at the effect of camel milk on ASD were reported (see Table 2) [13,17,41,42,43]. In one of those two studies a significant effect was reported, but not in the other. One additional original study was found that looked at the effect of camel milk on ASD symptoms [52] (see Table 3). Here, positive effects of camel milk on the CARS and on the Social Responsiveness Scale (SRS) were shown, especially for raw camel milk.

Two original studies were found in which the effect of tea containing GABA and L-theanine on ASD symptoms [53] and the relation between sugar based beverage consumption and ASD symptoms [54] were investigated (see Table 3). Five out of nine participants improved on ASD symptoms while consuming the tea containing GABA and L-theanine compared to the placebo tea [53]. The adjusted odds ratio for the Clancy Autism Behavior Scale increased with increasing sugar-based beverage consumption [54].

As there were fewer than five original studies related to the Chanyi diet, camel milk, tea containing GABA and L-Theanine, and sugar-based beverage consumption for ASD symptoms, no conclusions can be drawn on the effects of these diets or specific food products on ASD symptoms in childhood.

### 3.2. Fatty Acids

The five meta-analyses [20,55,56,57,58] and ten systematic reviews [13,17,41,42,43,59,60,61,62,63] focusing on the effect of N-3 fatty acid, N-6 or polyunsaturated fatty acid (PUFA) supplementation on ASD symptoms included a total of eleven unique studies (see Table 4).

In four of the five meta-analyses some small positive significant effects of fatty acid supplementation on ASD symptoms were shown [20,55,56,58], although the domains on which a beneficial effect was shown differed: anxiety [20], language [58], social-autistic [58], core symptoms of ASD [58], associated symptoms of ASD [58], hyperactivity [56], lethargy [55,56], stereotypical behavior [56] and daily living [55] (see Table 4). However, in two of these meta-analyses also some detrimental effects of fatty acid supplementation were shown. De Crescenzo showed a detrimental effect on sleep quality [20], and Horvath showed a detrimental effect on externalizing behavior and social skill [55]. In none of the meta-analyses evidence for heterogeneity or publication bias was found [20,55,56,58]. In one meta-analysis with only two RCTs no significant effect of N-3 fatty acids supplementation on ASD was shown [57]. Note that in the vast majority of studies included in the meta-analyses a rather limited number of participants was included (i.e., varying between 9 and 77 participants).

The systematic reviews in which the focus was on supplementation with N-3, N-6 fatty acids or PUFA to improve ASD symptoms showed mixed results. Four systematic reviews reported no effect (i.e., >2/3 of included studies showed no effect) of N-3 and/or N-6 fatty acid supplementation on ASD symptoms [17,41,43,60,61,62], two showed mixed effects [13,59] (i.e., <2/3 studies results in same direction), and two showed beneficial effects (i.e., >2/3 of included studies showed beneficial effects) [42,63].

In addition, two original studies were found (see Table 5). In both studies, significant beneficial effects of N-3 fatty acid supplementation on ASD symptoms were shown [64,65].

As fewer than 2/3 of studies showed an effect in the same direction, no conclusion can be drawn about the effect of N-3, N-6 fatty acid or PUFA supplementation on ASD symptoms during childhood.

### 3.3. Micronutrients

One meta-analysis [58] and 14 systematic reviews [13,17,21,41,42,43,60,61,66,67,68,69,70,71] were found in which the relationship between micronutrients and ASD was studied (see Table 6). In seven systematic reviews the focus was on specific micronutrients, namely: vitamin D [21,66,67], vitamin B6 in combination with magnesium [68,69,70], or L-carnitine [71]. In one meta-analysis and seven systematic reviews [13,17,41,42,43,60,61], several micronutrients (including vitamin B12, vitamin B6, vitamin C) and other nutrients, such as amino acids, inositol and carnitine were discussed.

#### 3.3.1. Vitamin D

Five systematic reviews were found, studying the relation between vitamin D and ASD [17,21,61,66,67] (see Table 6). The systematic reviews included a total of five unique studies in which the effect of vitamin D supplementation on ASD symptoms was investigated were included. Note that one of these five studies has been retracted due to concerns regarding the credibility of the data [72]. Two of the four remaining studies showed beneficial effects of vitamin D supplementation on the Aberrant Behavior Checklist Scale (ABC) score, and ABC and CARS respectively. One study did not show differential change of ASD measurement scores for the supplemented and non-supplemented group. Lastly, one study only showed an improvement on the self-care subscale of the Developmental Disability-Children’s Global Assessment Scale. However, these four studies were of poor quality (i.e., many were open label studies and included few participants).

In addition, five original studies (reported in seven manuscripts), in which the relation between vitamin D and ASD symptoms was investigated, were found [73,74,75,76,77,78,79] (see Table 7). In two larger scale RCTs, significant effects of vitamin D supplementation on respectively irritability and hyperactivity [76,77,78] and on stereotypical behavior were found [79]. In one cohort and one case control study, no significant associations were found between vitamin D supplementation at 2.5 years of age and ASD diagnosis at a mean age of 5.1 years [73]. In addition, no difference was found in the use of vitamin D drops during infancy between children with ASD and children without ASD [74].

As fewer than 2/3 of studies showed an effect in the same direction, no conclusion can be drawn about the effect of vitamin D supplementation on ASD symptoms during childhood.

#### 3.3.2. Vitamin B6 + Magnesium

Six systematic reviews in which the effect of vitamin B6 supplementation with or without magnesium on ASD symptoms was investigated [17,41,60,68,69,70] (see Table 6) were found. In these systematic reviews a total of five unique studies in which the effect of vitamin B6 in combination with magnesium on ASD symptoms was studied were included. Additionally, one study which looked at vitamin B6 and magnesium both separately and combined was included [89], and lastly one study in which only the effect of vitamin B6 was studied was included [90].

For the six studies in which the combined effect of vitamin B6 and magnesium was investigated, a beneficial effect was shown in four studies and no effect in the two others. For the two studies in which vitamin B6 was supplemented alone, a positive effect was found in one study and no significant effect in the other study. It should be mentioned that the number of participants included in the studies was very small (i.e., 37 participants in the largest study), and the exact outcome measures were unclear.

No conclusion can be drawn on the influence of vitamin B6, alone or in combination with magnesium on ASD symptoms, as fewer than 2/3 of studies showed an effect in the same direction.

#### 3.3.3. Iron

One original study was found in which the effect of ferrous sulfate supplementation in children with ASD, insomnia and low iron stores was investigated [85]. No differences in changes in ASD behavioral measures (i.e., Aberrant Behavior Checklist irritability scale; Swanson, Nolan, and Pelham-IV; Repetitive Behavior Scale-Revised) was shown after supplementation between the ferrous sulfate group and the control group.

No conclusion can be drawn about the effect of iron supplementation on ASD symptoms in childhood due to a too-limited number of studies (i.e., <5 studies).

#### 3.3.4. Folic Acid

In two systematic reviews one unique study was found that investigated the effect of folic acid supplementation [17,61]. In this study verbal communication, daily living skills, irritability, lethargy, stereotyped behavior hyperactivity, inappropriate speech, total score on the ABC, and internalizing problems improved more in the folic acid supplementation group than in the placebo group.

Additionally, one systematic review reported one trial in which the effect of folic acid in combination with vitamin B12 was investigated [41]. In this study, significant improvements on Vineland subscales were shown. Lastly, two orignal folate supplementation studies were found [81,82] (see Table 7). These studies, however, were of poor quality and did not show unambiguous results.

No conclusion can be drawn about the effect of folic acid supplementation for ASD symptoms due to a too-limited number of studies (i.e., <5 studies).

#### 3.3.5. Zinc

We found one original study in which the effect of daily zinc supplementation on CARS score was investigated [88]. In this study, a significant decrease in CARS score was shown. However, no control group was included in this study.

No conclusion can be drawn about the influence of zinc supplementation on ASD symptoms in childhood due to a too-limited number of studies (i.e., <5 studies).

#### 3.3.6. Vitamin B12

Seven systematic reviews identified a total of two unique studies in which the influence of vitamin B12 on ASD symptoms was investigated [13,17,41,42,43,60,61] (see Table 6). In both studies an improvement in Clinical Global Impression was reported (probably only significant in one study).

As noted earlier, one systematic review identified one trial in which the effect of combined vitamin B12 and folic acid was investigated. This study did show some beneficial improvements on Vineland sub scales [41]. 

No conclusion can be drawn about the effect of vitamin B12 alone or in combination with folic acid on ASD symptoms during childhood due to a too-limited number of studies (i.e., <5 studies).

#### 3.3.7. Amino Acids

In one systematic review seven studies in which the influence of amino acids on ASD symptoms was studied, were included [60] (see Table 6). Four of the seven studies included in this systematic review showed a positive effect of amino acid supplementation on ASD symptoms and especially irritability. It is, however, important to note that different amino acids were used in the studies; in four studies N-acetylcysteine was supplemented, in two d-Cycloserine and in one *N*,*N*-dimethylglycine.

No conclusion can be drawn about the effect of amino acid supplementation on ASD symptoms during childhood due to a too-limited number of studies (i.e., <5 studies) for each specific amino acid.

#### 3.3.8. L-Carnitine and L-Carnosine

In three systematic reviews, a total of three studies that studied the influence of L-carnitine on ASD were included [13,42,71] (see Table 6). All three studies showed a significant favorable effect of L-carnitine supplementation on ASD outcome measures such as CARS and ABC. In addition, Brondino et al. reported one study on the influence of L-carnosine on ASD symptoms [41]. This study showed a significant beneficial effect of L-carnosine supplementation on the Gilliam Autism Rating Scale (GARS) score when comparing the intervention group with the placebo group.

We lastly located one original study reporting on the influence of 500 mg of L-carnosine per day on GARS score; in this study no significant change in autism severity was shown after supplementation [87].

No conclusion can be drawn about the effect of L-carnitine or L-carnosine supplementation on ASD symptoms during childhood due to a too-limited number of studies (i.e., <5 studies).

#### 3.3.9. Multivitamins

One meta-analysis and three systematic reviews included a total of two unique studies in which the effect of multivitamin supplementation on ASD symptoms was investigated [41,43,58,68]. In one of the two studies beneficial effects of supplementation on ASD symptoms were shown. The other study did not show significant beneficial effects on ASD symptoms.

No conclusion can be drawn about the effect of multivitamin supplementation on ASD symptoms during childhood due to a too-limited number of studies (i.e., <5 studies).

#### 3.3.10. Vitamin A

One study in which the focus was on vitamin A supplementation was found in a systematic review, another study was located as an original study [43,80] (see Table 6 and Table 7). Only one of these studies [43] found a significant improvement in the CARS score and Diagnostic and Statistical Manual of Mental Disorders (DSM) criteria.

No conclusion can be drawn about the effect of vitamin A supplementation on ASD symptoms during childhood due to a too-limited number of studies (i.e., <5 studies).

#### 3.3.11. Other Nutrients

In the systematic reviews, a single study was included that looked at the influence of vitamin C [41,60], inositol [60], and flavonoids [41], see Table 6.

In one original study the relation between thiamine deficiency early in life and scores on ASD questionnaires later in life was investigated [83] and in another original study the effect of rerum (a supplement consisting of chondroitin sulphate, vitamin D3 and oleic acid) on ASD was investigated [84], see Table 7. Lastly, in one original study a personalized treatment regime to correct nutritional derangements and its effect of CARS was investigated [86].

No conclusion can be drawn about the effect of vitamin C, inositol, flavonoid, multivitamin, thiamine deficiency, rerum or a personalized nutritional program on ASD symptoms during childhood due to a too-limited number of studies (i.e., <5 studies).

## 4. Discussion

The goal of the current scoping review was to summarize the scientific literature, including identification of knowledge gaps, regarding the relation between nutrition and ASD symptoms in childhood. The current review shows the availability of studies in which therapeutic diets, specific food products, fatty acids and micronutrients for the treatment of childhood ASD symptoms were investigated. There were, however, no studies available on the relation between dietary patterns (i.e., the habitual total food intake) or macronutrients and childhood ASD symptoms. In addition, there was a large variability in nutritional aspects under study, size and quality of studies, and thereforewe could thus not draw conclusions for any of the therapeutic diets, specific food products, fatty acids or micronutrients. The lack of consistent findings derived from methodologically strong studies makes it impossible to formulate practical nutrition guidelines for the treatment of childhood ASD.

Although 98 studies were available, the focus of the majority of studies was on a few specific nutritional factors. For example, there were 28 studies (29%) with a focus on the GFCF diet and 13 with a focus on fatty acids (13%). Most nutritional factors were investigated in <5 studies (e.g., ketogenic diet, iron, folic acid and zinc), and therefore, no conclusions could be drawn about those specific nutritional factors. It is, thus, clear that there are major gaps in knowledge regarding the relation between childhood nutrition and ASD symptoms during childhood. Among others, the relation between dietary patterns and ASD symptoms, but also the causality between supplementation of various nutrients (including iron, zinc and magnesium) and ASD symptoms, remains unclear.

Simultaneously, the amount of studies available on the relation between nutrition and ASD symptoms during childhood suggests there is much interest in this topic. However, many of the studies were of poor methodological quality, which limits the evidence base for the relation between nutrition and ASD symptoms during childhood. This is even true for nutritional factors such as the GFCF diet and fatty acids, which have been studied in >5 studies. However, as the studies were of such limited methodological quality, no firm conclusions could be drawn.

The methodological quality of studies was limited by a number of factors, which should be considered for future studies. First, although many of the included studies only included children with a clinician-based diagnosis of ASD using standardized tools such as the DSM criteria, International Statistical Classification of Diseases and Related Health Problems (ICD) criteria, Autism Diagnostic Interview-Revised (ADI-R) or Autism Diagnostic Observation Schedule (ADOS), some studies did not report how the diagnosis was made or utilized other tools which might not be suitable for diagnosis such as CARS (for on overview of the tools used see Appendix A). It is of course important that to assess the effect of nutrition factors on ASD in children, the actual participants be children with clinician diagnosed ASD. While ASD symptoms were mostly assessed by the use of an ASD measurement, a multitude of instruments were used, and often multiple ASD measurement instruments were used within a single study (for overview of used measurement instruments see Appendix A). Not all ASD measurement instruments used measure the same concept, nor do they necessarily measure core-features of ASD. For example, the ADOS, CARS and SRS were developed as autism diagnostic tools and measure core-features of ASD, but many other scales measure associated symptoms, such as behavioral problems (i.e., ABC), adaptive behavior (Vineland Adaptive Behavior Scale (VABS)) or clinical global impression (CGI). If a scale does not measure a core symptom of ASD, the symptom might not be present in all participants included in the trial and the results of the trial are, thus, not valid. According to the European Medicine Agency, trials should first demonstrate efficacy on at least one core symptom of ASD. Only then, efficacy on other associated symptoms may be claimed [91]. Additionally, many of the studies utilized parent-reported outcomes, mostly as standalone outcome, but in some cases in combination with clinician-reported outcomes, for the assessment of the child’s ASD symptoms (for overview of used measurement instruments see Appendix A). And even though the agreement on the presence or absence of ASD symptoms is moderate to low between clinicians and parents, there are indications that parents and clinician do differ in their evaluation of specific behaviors [92]. It thus important that both clinician and parent-reports are included in studies. However, using multiple ASD measurements and, with that, analyzing multiple outcomes can lead to Type 1 errors if no correction for multiple testing is applied. Finally, many of the studies included very few participants and utilized an open or single blind design, which both lead to the introduction of biases into the study.

Future studies should use more rigorous methodology. Researchers should, for example, utilize validated instruments that measure core-features of ASD, predetermine one primary clinically relevant endpoint and correct for multiple testing if using multiple outcomes. Additionally, studies should include a sufficient number of participants, and utilize double blind research designs if possible (e.g., for supplements). Moreover, it is advisable to choose comparable designs (e.g., form of supplementation, measuring instruments, etc.), so comparisons between studies is easier, allowing more reliable pooling of data in meta-analyses and systematic reviews.

It has been noted that ASD is a heterogeneous disorder in its clinical representation, severity, and most probably also in its etiology [8]. Additionally, psychopathologic co-morbidities such as anxiety, depression, ADHD and intellectual disabilities are often present [93]. Taking these facts into consideration, it seems very likely that there is not a one-size-fits-all treatment for every person with ASD, and personalization is most likely needed. It might be that there are specific subgroups with certain biochemical or clinical features in which certain dietary interventions are specifically effective. To determine this, more research is needed and baseline characteristics such as baseline severity of ASD, comorbidities and demographics should be reported to reduce the heterogeneity of subgroups [94] and to enable elucidation of personalized treatment approached.

Furthermore, many of the included studies focus on the effect of isolated nutrients on ASD. A combination of nutrients may be more likely to evoke an effect than one isolated nutrient [95,96], because it matches the physiological requirement of the body better by representing the array of nutrients which are present in natural food and human diets. It would thus also be advisable to investigate the effect of combinations of nutrients. Additionally, others have postulated that dietary patterns may be more predictive of disease risk than single nutrients [97]. To the best of our knowledge, there are no studies available investigating the relation between dietary patterns and ASD. This relation does warrant research.

The main strength of the current review is its broad scope, investigating a wide range of nutrition factors related to ASD in children. A comprehensive search strategy was used, and rigorous methods were applied for ranking the quality of studies.

This review also has a number of limitations. We utilized a broad search strategy, as we wanted to include a broad range of studies investigating the relation between nutrition in relation to ASD in childhood. However, even with this broad search strategy it might be the case that we missed studies. Although, as we used a conservative number of studies (i.e., a minimum of five studies) with an effect in the same direction to draw a conclusion, it does not seem likely that studies that we missed could have changed the final conclusions. Additionally, this scoping review was a review of meta-analyses/systematic reviews and we chose to summarize the results in a qualitative manner. This is a common method in the execution of reviews of meta-analyses/systematic reviews. However, its main disadvantage is that original studies are counted multiple time (i.e., if an original study is included in multiple meta-analyses/systematic reviews). Consequently, the results of some studies were given more weight in the strength of evidence allocation. In future analyses this should be corrected for. In classifying the strength of the evidence (see Table 1), we took the strength of research designs as a starting point. For meta-analyses and systematic reviews, we did not consider their quality in the strength of evidence, which makes further refinement in the strength of evidence impossible.

## 5. Conclusions

The evidence for the relation between dietary patterns, therapeutic diets, and nutrients and ASD symptoms in childhood is generally insufficient and research suffers from methodological limitations. The relatively high prevalence and high social and personal costs associated with ASD in childhood justifies more research into the potential relation between nutrition and ASD symptoms in childhood. Future research should be methodologically sound, additionally studies on the relation between dietary patterns and ASD symptoms in childhood are needed. Considering, the results of this review, no clinical or practical recommendations regarding nutrition for ASD symptoms in childhood can be made.

## Figures and Tables

**Table 1 nutrients-14-01389-t001:** Criteria for strength of evidence and wording used in text, translated and adapted from [22].

Qualification and Description	Wording Strength of Evidence Used in Text
**Very Strong**	
Meta-analyses of studies with a strong research design (RCT and prospective cohorts) and ≥5 studies in total *	Very strong evidence for
**Strong**	
Systematic reviews of studies with a strong research design and ≥5 studies in total	Strong evidence for
**Moderately Strong**	
Large and well executed RCT Prospective cohorts Meta-analyses and systematic reviews of studies with strong and weak research designs (RCT, prospective cohorts, cross-sectional or case control) Meta-analyses of studies with a strong research design (RCT and prospective cohorts) and < 5 studies in total. *	Moderate strong evidence for
**Weak**	
Meta-analyses and systematic reviews of studies with a weak research designs (cross-sectional or intervention without control group) Non-randomized trials	Possible association/effect based on studies with a weak research design No proof for association/effect based on studies with a weak research design
**Very Weak**	
Cross-sectional studies Case-control studies Intervention without control group Open label trials Small and/or poorly executed RCT Meta-analyses and systematic reviews of mixed studies (i.e., different outcomes, different exposures and different designs).	Possible association/effect based on studies with a very weak research design No proof for association/effect based on studies with a very weak research design

* Meta-analyses of studies with a strong research design but high heterogeneity (I^2^ > 75%), where no attempt was made to explain the heterogeneity, were placed one level lower in terms of strength of evidence.

**Table 2 nutrients-14-01389-t002:** Meta-analyses and systematic reviews concerning therapeutic diets and specific food products for ASD.

Type of Paper	Diet	Outcome	Design(s) of Orginal Studies	Effects	Final Conclusion
Gluten and/or casein free diet					
Piwowarczyk 2018 SR [36]	Gluten and casein free diet	Behavior and ASD related outcomes	6 RCT	2/6 studies showed sig. differences in ASD core symptoms 2/6 studies showed sig. differences in sub domains	0
Mari-Bauset 2014 SR [31]	Gluten and/or casein free diet	ASD behavioral symptoms or biomedical symptoms	4 RCT, 6 trials	8/10 studies showed improvement 2/10 studies did not show sig. behavioral improvement	+
Hurwitz 2013 SR [30]	Gluten and casein free diet or gluten free diet	Behavior or developmental outcomes	4 RCT	2/4 studies showed a sig. positive effect	+/−
Buie 2013 SR [28]	Gluten and/or casein free diet	Autism	1 RCT, 4 trials	4/6 studies showed behavioral improvement	+
Mulloy 2010 + 2011 SR [34,35]	Diet that reduced/removed casein and/or gluten	Variables related to the improvement of ASD symptoms	4 RCT, 6 trials	6/10 studies showed positive effects of diet 2/10 studies showed no sig. effects 2/10 studies showed mixed effects	+/−
Millward 2008 + 2004 SR [32,33]	Gluten and/or casein free diet	Behavioral observations and standardized assessment of autistic behavior, communication and language.	2 RCT	1/2 studies showed positive effects of diet on autistic characteristics, communication and interaction and social isolation	+/−
Christison 2006 SR [29]	Gluten and/or casein elimination diet	Clinical measures in children with ASD	1 RCT, 5 trials	6/6 studies showed improvements	+
Ketogenic diet					
Kraeuter 2020 SR [39]	Ketogenic diet	ASD	1 RCT, 2 trials	3/3 studies showed improvement on some of the measures used.	+
Bostock 2017 SR [37]	Ketogenic diet	ASD	1 trial	40% of participants dropped out, but of the remaining participants two showed sig. improvements on the CARS, the rest showed mild to medium improvement	NA
Castro 2015 SR [38]	Ketogenic diet	Behavioral symptoms	2 trials	2/2 studies all participants showed improved scores on the CARS (unclear whether these improvements were sig.)	+/−
Different diets					
Monteiro 2020 SR [43]	(I) Gluten and/or casein elimination diet (II) Chanyi diet ^1^ (III) Camel milk	Behavioral symptoms of ASD	(I) 8 RCT, 1 trial (II) 1 RCT (III) 1 RCT	(I) Most studies did not show statistical improvement of clinical symptoms of ASD. 3 studies showed non-significant improved communication, stereotype movements, aggressiveness, and signs of ADHD. (II) Sig. improvement in several behavioral symptoms typical of ASD. (III) Sig. Improvement in communication and cognition.	NA
Gogou 2018 SR [40]	(I) Ketogenic diet (II) Gluten and casein free diet (III) Gluten free and adapted ketogenic diet	Not specified (different measures reported)	(I) 1 trial (II) 3 RCT, 1 trial (III) 1 trial	(I) 10/30 participants sig. or minor improvement (II) 2/4 studies showed sig. positive effects on clinical ASD aspects (III) Only improvement on social affect score	NA
Li 2017 SR [17]	(I) Gluten and casein free diet (II) Camel milk	Core symptoms of ASD	(I) 4 RCT, 1 trial (II) 2 RCT	(I) 4/5 studies showed no sig. differences. 1 study showed positive effect on psychotic behavior (II) Sig. improvement in CARS score in raw camel milk group	NA
Williamson 2017 SR [13]	(I) Gluten and/or casein elimination diet (II) Camel milk	Core symptoms and related symptoms of ASD	(I) 6 RCT (II) 1 RCT	(I) 2/6 studies showed improvement with diet on sub scales, 1/6 studies improvement on scale only after 12 months on the diet not after 24 months on the diet (II) No sig. difference in CARS score between camel milk and cow milk group.	NA
Sathe 2017 SR [42]	(I) Gluten and/or casein elimination diet (II) Camel milk (III) Gluten free diet (IV) Gluten and dairy free diet	ASD	(I) 4 RCT (II) 1 RCT (III) 1 RCT (IV) 1 RCT	(I) 3/4 studies no sig. effects of diet, 1/4 studies improvement only after 12 months on the diet, not after 24 months (II) No sig. difference in CARS score between camel milk and cow milk group. (III) Sig. improvement in stereotypical behavior and communication in gluten free group compared to control group (IV) No group differences between diet group and control group for challenging behavior.	NA
Brondino 2015 SR [41]	(I) Gluten and/or casein elimination diet (II) Ketogenic diet (III) Chanyi diet ^1^ (IV) Camel milk	Core symptoms of ASD	(I) 4 RCT (II) 1 trial (III) 1 RCT (IV) 2 RCT	(I) 2/4 sig. improvement of diet (II) Improvement due to diet (III) Sig. improvement in ATEC in the experimental group (IV) 2/2 sig. improvement on CARS	NA

+ = 2/3 or more of the included studies found a positive effect or association, − = 2/3 or more of the included studies found a negative effect or association, 0 = 2/3 or more of the included studies showed no effect or association was shown, +/− = the results in a systematic review or meta-analyses were mixed (i.e., <2/3 studies in the same direction), and there were at least five studies in which the effect/association was studied, ADHD = attention, deficit (hyperactivity) disorder, ASD = autism spectrum disorder, ATEC = Autism Treatment Evaluation Checklist, CARS = Childhood Autism Rating Scale, NA = not appropriate, RCT = randomized controlled trial, sig.= significant, SR = systematic review. ^1^ Chanyi diet is a traditional Chinese diet in which the intake of some foods which cause “internal heat” are limited.

**Table 3 nutrients-14-01389-t003:** Studies concerning therapeutic diets and specific food products for children with ASD, that are not included in the meta-analyses and systematic reviews in Table 2.

Author, Year	Study Design	Population	Age Average (SD) Range	Intervention	Outcome Measure	Result	Quality
González-Domenech 2020 [44]	Trial	ASD *n* = 37	8.9 years (4.0)	6 months GFCF diet, 6 months normal diet, randomized cross-over	ATEC, ABC, ERC	Non sig. decrease in the ATEC, ERC and ABS score after the GFCF diet was found, with very small effect sizes.	Moderate
Piwowarczyk 2020 [45]	Trial	ASD or autistic disorder *n* = 58	48 months (11)	8 week GFD run-in, either GFD or GD 6 months.	ADOS-II, SCQ, ASRS, VABS	No sig. difference between GFD and GD group on ADOS at baseline at 6 month follow up.	Moderate
González-Domenech 2019 [46]	Trial	ASD *n* = 28	8.1 years (3.9)	3 months GFCF diet, 3 months normal diet, randomized cross-over	ATEC, ABC, ERC	A non-significant decrease in ATEC scores after the GFCF diet was found.	Weak
Hafid 2018 [47]	Trial	ASD *n* = 20	6–12 years	GFCF diet, 1 year	CARS	12 participants had a substantial decrease in CARS score after 12 months.	Moderate
Pennesi 2012 [48]	CS	ASD *n* = 387	‘children’	Questionnaire on the implementation of the GFCF diet	Change in autism related symptoms and behavior	Parents who eliminated all gluten and/or casein reported bigger improvement in ASD behavior, physiological symptoms and social behavior after the start of the diet than those that did not eliminate everything. Children with food allergies, food sensitivity and gastro-intestinal problems showed a bigger improvement with GFCF than children without those problems.	NA
Harris 2012 [49]	CS	ASD *n* = 13	9 years (1.9) Range 5–12 years	FFQ to determine adherence to the GFCF diet	CARS	There was no sig. Correlation between adherence to the GFCF diet and CARS score. 100% of parents reported behavioral improvement of their child after starting the GFCF diet.	NA
Amin 2011 [50]	Trial	ASD *n* = 42	50.6 months (11.38)	GFCF diet for 6 months	CARS	There was statistical sig. Improvement in the average total CARS score, children without dermorfine in their blood had sig. improvement on 8/9 sub scales, children with dermorfine on 6/9 sub scales.	Weak
Nazni 2008 [51]	Trial	ASD *n* = 30	Range 3–11 years	CF, GF or GFCF diet advise 2 months	Behavior as noted by parents	In all three diet group the children showed behavioral improvement.	Weak
Al-Ayadhi 2015 [52]	RCT	ASD *n*= 65	7.8 years Range 2–12 years	500 mL raw camel milk, 500 mL boiled camel milk or 500 mL cow milk (placebo) per day for 2 weeks	CARS, SRS and ATEC	Sig. decrease in CARS for both the raw and boiled camel milk group. Children who received raw camel milk showed sig. Reduction on 3/5 sub scales of SRS, boiled camel milk on 1/5 sub scales and cow milk showed no sig. reduction. On the ATEC there was only a sig. reduction on 1 sub scale for the boiled camel milk group.	Weak
Hannant 2019 [53]	RCT	ASD *n* = 9	11.58 years (0.58)	Three teas with varying amount of GABA and L-Theanine. Each tea 2 weeks, with 1 week wash-out between teas.	ADOS-II and ASRS	5/9 students showed improved symptoms related to DSM-5 criteria with GABA tea and with L-Theanine tea.	Moderate
Geng 2020 [54]	CS	NA *n* = 27,200	3–6 years	Questionnaire to assess sugar based beverage consumption	CABS	Adjusted ORs (95%CI) for CABS increased across the SSB categories 1.00 (<1 time per day), 1.17 * (1 time per day) and 1.57 * (>2 times per day).	NA

* = significant results, ABC = The Aberrant Behavior Checklist Scale, ADOS = Autism Diagnostic Observation Schedule, ASD = autism spectrum disorder, ASRS = Autism Spectrum Rating Scale, ATEC = Autism Treatment Evaluation Test questionnaire, CABS = Clancy Autism Behavior Scale, CARS = Childhood Autism Rating Scale, CF = casein free, CS = cross-sectional, DIPAB = Diagnosis of Psychotic Behavior in Children, ERC = Evaluation Resumé du Comportement, in French = The Behavioral Summarized Evaluation, FFQ = food frequency questionnaire, GD = gluten containing diet, GFCF = gluten free casein free diet, GFD = gluten free diet, NA = not appropriate, RCT = randomized controlled trial, SCQ = Social Communication Questionnaire, sig. = significant, SRS = Social Responsiveness Scale, VABS = Vineland Adaptive Behavior Scale.

**Table 4 nutrients-14-01389-t004:** Meta-analyses and systematic reviews concerning the effect of N-3, N-6 or PUFA supplementation on ASD.

Type of Paper	Nutritional Factor	Outcome	Design(s) of Orginal Studies	Effects	Final Conclusion
De Crescenzo 2020 MA [20]	PUFA supplementation	Outcomes determined to be highly relevant for children and adolescent with ASD, determined by expert panel.	8 RCT ^1^	PUFAs were superior compared to placebo in reducing anxiety in individuals with ASD (SMD = −1.01 *, very low certainty of evidence). PUFAs worsened quality of sleep compared to a healthy diet (SMD = 1.11 *, very low certainty of evidence). PUFAs were not better than placebo in reducing aggression, hyperactivity, adaptive functioning, irritability, restricted and repetitive interests and behaviors and communication.	+/−
Fraguas 2019 MA [58]	Omega-3 PUFA supplementation	ASD	6 RCT ^2^	Omega-3 supplementation was sig. more effective than placebo in treating the following symptoms and/or function groups: language (g = 0.313 *) and social-autistic (g = 0.311 *) And sig. more effective for the following clinical domains: core symptoms (g = 0.268 *) and associated symptoms (g = 0.276 *).	+
Cheng 2017 MA [56]	N-3 fatty acids supplementation	Change in ASD severity scale or change in secondary ASD behavioral symptoms	5 RCT ^3^	Only small sig. effects on the ABC: hyperactivity (g = −0.348 *), lethargy (g = −0.447 *) en stereotypical behavior (g = −0.404 *).	+
Horvath 2017 MA [55]	N-3 fatty acids supplementation	ASD symptoms	5 RCT	For most used measuring instruments/scales no sig. effects. Studies that used ABC sig. improvement on lethargy in n-3 group (pooled MD = 1.98 *). Studies that used the BASC sig. worsening of externalizing behavior (pooled MD = −6.22 *) and social skill in the n-3 group. One study showed an improvement on the VABS daily living component for those in the n-3 group (MD = 6.2 *).	+/−
James 2011 MA [57]	N-3 fatty acids supplementation	Improvement in social interaction, communication or stereotypical behavior.	2 RCT	No sig. improvements.	0
Monteiroa 2020 SR [43]	Omega-3 supplementation	Behavioral symptoms of ASD	4 RCT	No changes observed in patients	0
Agostoni 2017 SR [59]	N-3 fatty acids supplementation	Core symptoms of ASD	2 RCT	Every study showed improvement for the supplemented group on some (sub) scales but not all.	+/−
Gogou 2017 SR [60]	Fatty acids supplementation	Clinical parameters of ASD	4 RCT	4/4 studies no sig. effects.	0
Li 2017 SR [17,61] ^4^	N-3 fatty acids supplementation	Core symptoms of ASD	5 RCT	4/5 studies showed no sig. effects of supplementation. 1/5 studies showed sig. effects on some (sub) scales.	0
Sathe 2017 R [42]	N-3 fatty acids supplementation	ASD	4 RCT	3/4 studies showed improvement on one or more sub scales.	+
Williamson 2017 SR [13]	Omega-3 supplementation	Core symptoms or related symptoms of ASD	4 RCT	For most used instruments/scales no sig. effects. 2/4 studies had a sig. improved score for supplementation group in comparison to place group for some sub scales. 1/4 studies reported sig. better scores for placebo group in comparison to the active group on a sub scale.	+/−
Brondino 2015 SR [41]	Omega-3 fatty acid supplementation	Core symptoms of ASD	5 RCT	4/5 studies no sig. differences. 1/5 study 20/30 showed improvement	0
Roux 2015 SR [62]	N-3 fatty acids supplementation	Behavioral problems in children with ASD	5 RCT, 1 trial	3/6 studies improvement, but not sig. 2/6 studies no sig. improvement. 1/6 studies sig. improvement on the ATEC	0
Bent 2009 SR [63]	N-3 fatty acids supplementation	Core symptoms of ASD or related symptoms	1 RCT, 2 trials	3/3 studies showed improvement (1 trend, 2 unclear)	+

+ = 2/3 or more of the included studies found a positive effect or association, − = 2/3 or more of the included studies found a negative effect or association, 0 = 2/3 or more of the included studies showed no effect or association was shown, +/− = the results in a systematic review or meta-analyses were mixed (i.e., <2/3 studies in the same direction), and there were at least five studies in which the effect/association was studied, * = significant results, ABC = Aberrant Behavior Checklist, ASD = autism spectrum disorder, ATEC = Autism Treatment Evaluation Checklist, BASC = Behavior Assessment System for Children, MA = meta-analysis, MD = mean difference, PUFA = polyunsaturated fatty acid, RCT = randomized controlled trial, sig. = significant, SMD = standardized mean difference, SR = systematic review, VABS = Vineland Adaptive Behavior Scale; ^1^ One study was executed in only adolescents, this study is excluded here, but is included in the MA. ^2^ One study was executed in only adolescents, this study is excluded here, but is included in the MA. ^3^ In one study the average age was 14.6 years, this study is excluded here, but is included in the MA. ^4^ Both manuscripts reported the same study with the same results.

**Table 5 nutrients-14-01389-t005:** Studies concerning N-3, N-6 or PUFA supplementation for ASD, that are not included in the meta-analyses and systematic reviews in Table 4.

Author, Year	Study Design	Population	Age Avarage (sd) Range	Intervention	Outcome Measure	Result	Quality
Keim 2018 [64]	RCT	Prenatal + Increased score on ASD questionnaire	Median intervention: 30 months; Median placebo 25 months Range 18–38 months *n* = 31	706 mg n-3 FA (among others 338 mg EPA, 225 mg DHA), 280 mg n-6 FA en 306 mg n-9 FA Or placebo 90 days	PDDST-II, BITSEA, 2 Separate questions on divided attention and reacting to name	The intervention group had sig. larger improvement on BITSEA ASD scale than the placebo group (−0.71 standardized effect size, medium-large effect).	Strong
Ooi 2015 [65]	Trial	ASD	11.66 years (3.05) *n* = 41	840 mg DHA, 192 mg EPA, 66 mg AA, 144 mg GLA, 60 mg Vitamin E, 3 mg thyme oil No placebo group.12 weeks	SRS-P and CBCL	Sig. improvements on all scales of the SRS-P (social consciences, social cognition, social communication, social motivation, autistic manners and total score) Sig. improvement on two sub scales of CBLC (social problems and attentional problems)	Weak

AA = arachidonic acid, ASD = Autism Spectrum Disorder, BITSEA = Brief Infant Toddler Social and Emotional Assessment, CBCL = Child Behavior Checklist, DHA = docosahexaenoic acid, EPA = eicosapentaenoic acid, FA = fatty acid, GLA = gamma-linolenic acid, PDDST-II = Pervasive Developmental Disorders Screening Test II, RCT = randomized controlled trial, SRS-P = Social Responsiveness Scale Parent.

**Table 6 nutrients-14-01389-t006:** Meta-analyses and systematic reviews concerning the relationship between vitamins and minerals, and ASD.

Type of Paper	Nutritional Factor	Outcome	Design(s) of Orginal Studies	Effect	Final Conclusion
Vitamin D					
Focker 2017 SR [21]	Vit D	ASD	1 RCT, 2 trial	RCT showed no sig. effect of vitamin D supplementation, 3/3 trials showed a positive effect of vitamin D supplementation on respective symptom scores.	+
Gillberg 2017 SR [66]	Vit D	ASD	1 RCT, 2 trials	2/3 studies showed improvement in ASD symptoms.	+
Mazahery 2016 SR [67]	Vit D	ASD	1 RCT, 3 trials	2/4 studies showed positive effects on some (sub) scales, 2/4 studies showed no effect of supplementation.	+
Vitamin B6 + magnesium					
Nye 2002 SR [69]	Vit B6 + Magnesium	ASD	2 RCT	2/2 studies no sig. treatment effect/no difference between intervention and placebo.	0
Kleijnen 1991 SR [70]	Vit B6	ASD	1 trials, 3 unclear	2/2 studies that gave vitamin B6 in combination with magnesium showed some positive effect. 1/2 studies that only gave vitamin B6 showed no sig. effect.	+/−
L-carnitine					
Malaguar- nera 2019 SR [71]	L-carnitine	ASD	2 RCT, 1 trial	3/3 studies significant effects of supplementation	+
Multiple vitamins					
Fraguas 2019 MA [58]	Different vitamin/mineral supplementation trials in 1 meta-analysis ^1^	ASD	7 RCT	Vitamin supplementation was sig. more effective than placebo in treating the following symptoms and/or function groups: global severity (g = 0.464 *), language (g = 0.351 *), stereotypies, restricted and repetitive behaviors (g = 0.531 *), behavioral problems and impulsivity (g = 0.402), and hyperactivity and irritability (g = 0.426 *). And sig. more effective for the following clinical domains: core symptoms (g = 0.308 *), associated symptoms (g = 0.308 *) and clinical global impression (g = 0.403 *).	+
Monteirao 2020 SR [43]	(I) Multivitamin (II) Methyl B12 (III) Vit A	Behavioral symptoms of ASD	(I) 1 RCT (II) 1 RCT (III) 1 trial	(I) No sig. difference (II) Sig. improvement in typical autism symptoms (III) Sig. progress in several clinical symptoms	NA
Li 2017 SR [61]	(I) Vit B12 (II) Vit D3 (III) Folic acid	Core symptoms of ASD	(I) 1 RCT, 1 trial (II) 1 RCT (III) 1 RCT	(I) 2/2 studies improvement on the CGI-I score, but not on other measures. (II) Sig. improvements on all measures (ABC, CARS, ATEC, SRS) (III) Sig. improvement verbal communication and overall, VABS, ABC, ASQ, en BASC.	NA
Li 2017 SR [17]	(I) Vit B6 (+magnesium) (II) Methyl B12 (III) Vit D3 (IV) Folic acid	Symptoms of ASD	(I) 4 RCT (II) 2 RCT (III) 2 RCT (IV) 1 RCT	(I) 1/4 studies positive effect, 1/4 studies potential improvement on communication and general responsiveness, 2/4 studies no effect. (II) 2/2 studies improvement on the CGI-I score, but not on other measures. (III) 1 study sig. improvement on all measures (ABC, CARS, ATEC, SRS), 1 study only sig. improvement on self-scare on the DD-CGAS (not ABC, SCQ of SRS). (IV) Sig. improvement verbal communication, VABS, ABC, ASQ, en BASC.	NA
Gogou 2017 SR [60]	(I) Amino acids (II) Vit B6 (+magnesium) (III) Vit B12 (IV) Vit C (V) Inositol	Clinical aspects of ASD	(I) 7 RCT (II) 3 RCT (III) 1 RCT (IV) 1 RCT (V) 1 RCT	(I) 3/7 studies no positive effects, 4/7 positive effects, of which 3 with n-acetylcysteine on irritability. (II) 2/3 studies no positive effects, 1/3 studies positive effects. (III) Positive effects on CGI score in a sub group. (IV) Positive effects (V) No positive effects	NA
Sathe 2017 SR [42]	(I) Methyl B12 (II) L-carnitine	ASD	(I) 2 RCT (II) 2 RCT	(I) 1/2 studies showed improvement on CGI in supplementation group in comparison to placebo. (II) 1/2 studies showed improvement in the severity of symptoms in the supplementation group in comparison to placebo.	NA
Williamson 2017 SR [13]	(I) Methyl B12 (II) L-Carnitine	Core symptoms and related symptoms of ASD	(I) 2 RCT (II) 2 RCT	(I) 1/2 studies showed improvement on CGI in supplementation group, but few other differences. (II) 1/2 studies showed improvement in the severity of symptoms in the supplementation group in comparison to placebo.	NA
Brondino 2015 SR [41]	(I) Vit B6 + magnesium (II) Methyl B12 (III) Methyl B12 + folic acid (IV) Vit C (V) Multivitamins (VI) L-carnosine (VII) Flavonoids	Core symptoms of ASD	(I) 2 RCT (II) 1 RCT (III) 1 trial (IV) 1 RCT (V) 1 RCT (VI) 1 RCT (VII) 1 trial	(I) 2/2 studies no difference between groups (II) No sig. differences between groups (III) Improvement in VABS sub scales (IV) Sig. improvements in vitamin C supplementation group (V) Sig. improvement in irritability in supplementation group (VI) Sig. improvements in supplementation group on GARS (VII) Sig. improvements supplementation group on VABS and ABC	NA
Murza 2010 SR [68]	Multivitamin	ASD	1 RCT	No effect supplement on ASD symptoms (expressive and receptive language, general behavior, eye contact and sociability).	0

+ = 2/3 or more of the included studies found a positive effect or association, − = 2/3 or more of the included studies found a negative effect or association, 0 = 2/3 or more of the included studies showed no effect or association was shown, +/− = the results in a systematic review or meta-analyses were mixed (i.e., <2/3 studies in the same direction), and there were at least five studies in which the effect/association was studied, * significant result, ABC = Aberrant Behavior Checklist, ASQ = Autism Symptoms Questionnaire, ASD = autism spectrum disorder, ATEC = Autism Treatment Evaluation Checklist, BASC = Behavior Assessment Scale For Children, CARS = Childhood Autism Rating Scale, CGI(-I) = Clinical Global Impression Scale (of Improvement), DD-CGAS = Developmental Disabilities—Children’s Global Assessment Scale, GARS = Gilliam Autism Rating Scale, MA = meta-analyze, NA = not appropriate, RCT = randomized controlled trail, SCQ = Social Communication Questionnaire, sig. = significant, SR = systematic review, SRS = Social Responsiveness Scale, VABS = Vineland Adaptive Behavior Scale, vit = vitamin. ^1^ Studies included supplemented with: multivitamin supplement, a multivitamin and mineral supplement, vitamin B12, vitamin C, vitamin B6 in combination with magnesium, folinic acid, and vitamin B6; these studies were combined in one meta-analysis.

**Table 7 nutrients-14-01389-t007:** Studies concerning vitamin/mineral supplementation for ASD, that are not included in the meta-analyses and systematic reviews in Table 6.

Author, Year	Study Design	Population	Mean Age (SD)	Intervention	Outcome Measure	Result	Quality
Vitamin D							
Mazahery 2019, 2020 [76,77,78]	RCT	ASD diagnosis *n* = 111	5.5 yeas (1.3) Range: 2.5–8 years	Four groups: (I) vit D supplementation 2000 IU perday, (II) 722 mg DHA per day (III) vit D + DHA (IV) Placebo 12 months	SRS, ABC	The SRS-social awareness subscale improved sig. greater for the DHA group and the combined group compared to the placebo. There was a trend in greater improvement for the SRS social communicative functioning score for the combined group and the SRS total score for the DHA group both compared to placebo. Sig. greater reduction on ABC irritability and hyperactivity for vitamin D and DHA group and a trend for a greater reduction for combined group all compared to placebo. Sig. greater reduction on ABC lethargy for DHA group compared to placebo. Evidence for interaction between baseline inflammation marker and treatment response was shown, children with elevated inflammation benefited more from supplementation.	Moderate
Moradi 2020 [79]	RCT	ASD diagnosis + serum vitamin D <30 ng/mL *n*= 100	7.62 years (1.15)	Four groups: (I) perceptual motor exercises, (II) vit D3 supplementation 300 IU/kg/day, (III) perceptual motor exercises + vit D3 supplementation 300 IU/kg/day, (IV) placebo 3 months	GARS stereotypy behavior subscale	A sig. reduction in stereotypical behavior was shown in all three experimental groups, but not in the placebo group. Stereotypical behavior sig. improved in the combined group compared to the other three groups.	Weak
Ali 2018 [73]	P-CH	*n* = 2526 (blood) *n* = 3825 (supplementation)	Baseline: 2.5 years (1.6). Follow-up: 5.1 years (2.3)	vit D supplementation questionnaire at baseline	ASD diagnosis by physian	No sig. ass between vit D supplementation and development of ASD (r = 0.86)	NA
Bittker 2018 [74]	CC	ASD *n* = 1515	Cases: 7.3 years (2.9) Controls: 5.5 years (2.6)	Online questionnaire vit D drops as baby (duration and dose) and folic acid supplementation during pregnancy.	ASD diagnosis	Vit D drops (months × dose): OR = 0.982 (not sig.), aOR = 1.006 (not sig.). Folic acid: OR = 0.785 (not sig.), aOR = 1.054 (not sig.).	NA
Jia 2018 [75]	Trial	ASD *n* = 3	38, 19 en 48 months	Vit D supplementation ^1^	ABS and CARS	The score on the CARS and the ABC fluctuated with the vit. D serum levels.	NA
Vitamin A							
Liu 2017 [80]	Trial	ASD *n* = 64	62.5 months (16.34) Range 1–8 years	One time 200.000 IU vit A supplementation; no placebo. Follow-up after 6 months	SRS, CARS and ABC	No difference in ABC, CARS or SRS after vit A supplementation.	Weak
Folic acid							
Sun 2016 [81]	Trial	ASD diagnosis *n* = 66	Interv: 57.23 months (15.06), Controls: 51.75 months (12.72)	800μg folic acid per day No supplement 3 months ^2^	ATEC, CARS, ABC and PEP-3	5/25 of the reported (sub) scales showed sig. interaction between the intervention and education program.	Weak
Gillberg 1986 [82]	Trial	Autism + fragile x positive + intellectual disability *n* = 4	Range 6–14 years	Folic acidsupplementation 0.5 mg/kg/day (A) of Placebo (B) for 3 months in A-B-A of B-A-B design.	ABC and other ASD checklists and questionnaires	1 participants improved with folic acid, 1 participants showed no effect, 2 participants showed unclear effects.	Weak
Other nutrients							
Fattal-Valevski 2009 [83]	P-CH	ASD *n* = 40	Exposed: 31.8 months (4.1) Controls: 32.2 months (3.9) Range 24–39 months	>1 month exposed to infant nutrition without thiamin (vit B1) in first year of life but no neurological abnormalities at start of study or control group.	M-CHAT and CARS	No sig. differences for CARS or M-CHAT for the thiamin deficient group and the control group.	NA
Antonucci 2017 [84]	Trial	ASD *n* = 44	2.5–14 years	0.01 mL rerum per week = supplement that contains chondroïtinesulfaat, vit D3 and oleic acid. Some children received more based on body weight. 2 months	Adapted CGI-I	32/44 participants showed an improvement 28/32 participants showed medium to severe improvement.	Weak
Reynolds 2020 [85]	RCT	ASD + insomnia *n* = 20	Active: 6.0 years (3.1), range 2.0–9.9 years Placebo: 5.7 years (2.8) range 2.0–10.1 years	3 mg/kg/day ferrous sulfate or placebo	ABC, SNAP-IV, the RBS-R	No difference in changes in the behavioral measures.	Strong
Ramaekers 2019 [86]	Trial	Non-syndromic infantile autism *n* = 166	Active 1–15.9 years Reference: 1–16.8 years	Treatment protocol for each individual aimed at correcting nutritional derangements (deficient or excess) adapted every 3–4 months. High dose folonic acid for those with FRα antibodies (0.5–1 mg/kg/day) increase to 2 mg/kg/day if no effect after 6 months. Control group: nothing. 2 years	CARS	CARS score dropped sig. after treatment.	Weak
Mehrazad-Saber 2018 [87]	RCT	Autism diagnosis with sleep disorder *n* = 50	Active: 8.59 years (2.77) Placebo: 8.35 years (2.76)	500 mg L-carnosine or placebo. 2 months	GARS	No sig. change in autism severity from baseline. No sig. difference in autism severity between active and placebo group.	Weak
Meguid 2019 [88]	Trial	Autism diagnosis *n* = 30	3–8 years	Daily zinc supplementation equal to bodyweight + 15–20 mg. No placebo. 12 weeks	CARS	CARS score was sig. lower after supplementation.	Moderate

ABC = Aberrant Behavior Checklist, aOR = adjusted odds ratio, ASD = autism spectrum disorder, ATEC = Autism Evaluation Treatment Checklist, CARS = Childhood Autism Rating Scale, CC = case control, CGI-I = Clinical Global Impression Scale of Improvement, DHA = docosahexaenoic acid, GARS = Gilliam Autism Rating Scale, IU = international unit, M-CHAT = Modified Checklist for Autism in Toddlers, NA = not appropriate, OR = odds ratio, P-CH = prospective cohort, PEP = Psychoeducational Profile, RBS-R = Repetitive Behavior Scale Revised, RCT = randomized clinical trial, sig. = significant, SRS = Social Responsiveness Scale, SNAP-IV = Swanson, Nolan and Pelham IV, vit = vitamin,. ^1^ For 1 child it was mentioned that he received 150,000 IU vitamin D intermuscular once per month plus 800 IU per day orally, for the other two children it was only mentioned that they received vitamin D supplementation. ^2^ All participants also took part in a specific education program for children with ASD.

## Data Availability

Not applicable.

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
