# Peer review of "A Scoping Literature Review of the Relation between Nutrition and ASD Symptoms in Children"

_nutrients, 2022, doi:10.3390/nu14071389_

Round 1

Reviewer 1 Report

Dear Authors, 

Thank you very much for the opportunity to revise this important work that collects data about nutrition and ASD symptoms in 0-12 aged children and depicts a clearer picture about the topic. Despite its relevance there are some major issues that should be addressed before its official publication in Nutrients Journal.

This work has the strength to collect a wide range of works considering this topic, however, the information collected is not always accurately and analytically described, especially considering the ASD symptomatology and its measurement.

Different studies used different outcome measures that target different aspects of ASD symptoms. In addition to this, different measures are administered in different ways and should be treated and discussed separately. See more detailed comments below.

English needs moderate editing considering oversights and some structural changes in some phrases should be applied to add fluidity.

Abstract

  • I suggest the authors structure the abstract starting with a few words on theoretical background before stating the goals.

Introduction 

  • Line 56: I would underline more why psychological interventions might not be available for all the persons adding a few lines to better clarify this aspect and, therefore, collocate the nutrition interventions as an option. Otherwise, the link might be confused.

Materials and Methods

  • Line 192: In considering the strength of the studies I would suggest authors also consider how the diagnosis was conducted. A diagnosis conducted through standardized tools is more powerful and revealing than parents’ or teachers’ reports. I would consider and discuss this aspect more when dealing with results. Behaviors noted by parents might contain less objectivity and more representations that the parents have considering their children.

Results 

  • In general I think that a distinction should be made between effects found in standardized tests and effects found in parental reports. According to me, this is the main issue concerning this study.
  • The authors pointed out this aspect in the discussion as a limitation but it should be better incorporated into the study design. Apart from the differences in the variety of tools in targeting different areas of ASD symptoms, I think that at least one distinction should be made between standardized tools applied by professionals (e.g., ADOS) and standardized tools based on reports of familiar persons such as parents of teachers of the child (e.g. VABS). Even if this work is comprehensive, it may lack precision if ASD symptomatology is taken as a whole without any distinctions. The idea of spectrum gives us the possibility to understand how many differences are contained in these symptoms and they should be taken into consideration at least partially.

Discussion / Conclusion

  • In general, apart from the methodological analysis and gaps identified in literature, the clinical implications and the importance of nutrition should be better incorporated into discussion or conclusion sessions.

Reviewer 2 Report

Journal: Nutrients

 Manuscript number: nutrients-1609817

 Title: A scoping literature review of the relation between nutrition and ASD symptoms in children

 This scoping review synthesized meta-analyses and systematic reviews, and original studies that examined the relationship between nutrition and ASD symptoms in children. The manuscript contributes to the literature in the relevant research topic by a comprehensive scoping review of both systematic reviews/meta-analyses and original studies. The review examined a wide range of nutrients and dietary factors for ASD, which is a large amount of work. I have several comments described below. Addressing the following comments would strengthen the manuscript.

  • Introduction

In general, the introduction is well organized. However, I think the introduction could be improved a bit by providing some background information about the findings and gaps from existing studies such as systematic reviews and meta-analyses for the topic of interest. For example, what we have known, and what need to be studied. Information emphasizing the importance of this scoping review over previous systematic reviews and meta-analyses would be helpful.

  • Materials and Methods

Line 93-99: the database search for the literature should state the publication period of the articles.  

Page 3: lines 105-108: “If multiple meta-analyses or systematic reviews were available, a review of reviews was executed. ……. If no meta-analyses or systematic reviews were available, the results of the original studies were summarized.”

I suggest that this paragraph be removed. An acknowledgement of the availability of multiple meta-analyses or systematic reviews would be better since the knowledge of the previous systematic reviews was the basis for the present study.   

Inclusion and exclusion criteria

Lines 117-122:

Inclusion criteria: “(6) the manuscript was a meta-analysis, systematic review, or a peer-reviewed original study with a quantitative design.”

“Studies were excluded if the study (1) was a single-case study; (2) a cross-sectional study reporting on nutrition intake or nutritional status of children with ASD, or a case-control study……;”

It is unclear about the types of quantitative designs defined for original studies in the eligibility criteria. The author stated that “a case-control study and a cross-sectional study reporting on nutrition intake or nutritional status of children with ASD were excluded.” But in the Results section, some cross-sectional studies and case-control studies are presented (e.g., Pennesi 2012, Geng 2020 in Table 3; Bittker 2018 in Table 7). The study designs for original studies should be clearly explained in the inclusion and exclusion criteria, and whether the criteria also apply to the original studies synthesized in the systematic reviews/meta-analyses. It appears that in the review of the systematic reviews/meta-analyses (Table 2, Table 4 and Table 6), only experimental studies (trials, RCTs) were included in the synthesis (note that a few systematic reviews included both experimental studies and observational studies). But this was not clearly stated in the inclusion criteria.

In addition, the nutritional factors are also needed to be more clearly specified in the inclusion and exclusion criteria. It seems confused by saying “a cross-sectional study or a case-control study reporting on nutrition intake or nutritional status of children with ASD were excluded.” What specific nutrition intakes were considered not eligible for inclusion in the review? How did the author define dietary patterns? Some dietary intakes or food products usually constitute dietary patterns, such as fruits and vegetables, grain product, or other foods groups (e.g., fast food). 

Page 3: 1. Procedure

This section mainly described the data extraction. I would use “Data extraction” instead of “Procedure”.

 2. Analyses

Lines 171-172: “If no meta-analyses or systematic reviews were available, the results of original studies were summarized in a review of original studies.”

This sentence is not clear. It needs to be revised, e.g., changing it as “for those original studies that were not included in the meta-analyses or systematic reviews, and met the inclusion criteria, the results of these studies were summarized in a review of original studies.”

 Page 5: 2.4. Strength of evidence

The authors described the levels of strength of evidence, e.g., from “very strong, strong to insufficient evidence and no evidence”, and showed table 1 as reference table, but did not present the result indicating the level of strength of evidence either in the synthesis of systematic reviews and meta-analyses or the original studies. It should be noted that quality (risk of bias) assessment of individual studies is different from grading the overall strength of evidence (see the reference below and the related other papers). I would suggest the authors make edits for the sentence in line 193-194 to avoid generating confusion.

Balshem H, Helfand M, Schunemann HJ, Oxman AD, Kunz R, Brozek J, et al. GRADE guidelines: 3. Rating the quality of evidence. Journal of clinical epidemiology. 2011;64(4):401-6.

  • Results

For Tables:

For each study (align the text for main findings to the top of the cell), it is better to align also the text to the top of the cells in other columns such that the table results are more readable (e.g., Table 6).  

In Table 2, Table 4 and Table 6, “study design(s)” should be changed to “Design(s) of the original studies”.  

Notes for tables: in table 2, “R” in the first column for the study was noted as “randomized” but in table 4 and table 6, “R” was labeled as “review”. It would be better to indicate systematic review as “SR”, and show it consistently in the tables for the systematic reviews.

(5) Please add a note for P-CH in Table 7. Please also check all the notes below the tables to make sure the abbreviations correspond to that showed in the tables (e.g., DB, CO for Table 2, no CC for case control in Table 4 and Table 6).

Line 574: ADHD: please provide the full name of the mental disorder when it is first present in the manuscript.

Since the review included studies reported in Dutch or German, in the results section, it should  summarize whether there were studies included in Dutch or German.  

  • Discussion

Lines 518-519: remove the repeated content “In total, five meta-analyses, 29 systematic reviews and 27 original studies were included in the current review.”  

Lines 519-520: “In total, 72 unique studies were included in the previous meta-analyses and systematic reviews, amounting to 99 unique studies included in the current review.”

This sentence can be moved to the Results section.

Lines 591-592: “all nutrition factors”. “all” can be changed to “a wide rang of”.  Not all nutrition factors were examined in this study.

References:

References should be numbered in order of appearance in the text of the manuscript.

Round 2

Reviewer 1 Report

Dear Authors,  

Thank you for the opportunity to revise again this manuscript that collects data about nutrition and ASD symptoms in 0-12 aged children providing a clearer picture about the topic.

This work has important strengths and it benefited from several clarifications that have been provided. The all comments have been addressed and justified satisfactorily, for this, I recommend this manuscript for the official publication in Nutrients Journal.

Reviewer 2 Report

The authors have addressed all my comments. I do not have additional comments.